# Development of Waste-Based Alkali-Activated Cement Composites

**DOI:** 10.3390/ma14195815

**Published:** 2021-10-05

**Authors:** Adrienn Boros, Csilla Varga, Roland Prajda, Miklós Jakab, Tamás Korim

**Affiliations:** 1Department of Materials Engineering, Faculty of Engineering, University of Pannonia, H-8201 Veszprém, Hungary; prajdaroland28@gmail.com (R.P.); jakab.miklos@mk.uni-pannon.hu (M.J.); ktm042@almos.uni-pannon.hu (T.K.); 2Sustainability Solutions Research Lab, Faculty of Engineering, University of Pannonia, H-8201 Veszprém, Hungary; vcsilla@almos.uni-pannon.hu

**Keywords:** alkali-activated cement, composite materials, waste rubber, fibre reinforcement, mechanical properties, rubber surface treatment, cyclic loading, scanning electron microscopy, FT-IR spectroscopy, computed tomography

## Abstract

Nowadays, global warming and the ensuing climate change are one of the biggest problems for humanity, but environmental pollution and the low ratio of waste management and recycling are not negligible issues, either. By producing alkali-activated cements (AACs), it is possible to find an alternative way to handle the above-mentioned environmental problems. First, with a view to optimizing experimental parameters, metakaolin-based AACs were prepared, and in it, waste tire rubber was used as sand replacement (5–45 wt %). Insufficient wetting between the rubber particles and the matrix was corrected through different surface treatments of the rubber. For improving the mechanical/strength properties of the specimens, fibrous waste kaolin wool (0.5–1.5 wt %) was added to the AAC matrix. Considering the results of model experiments with metakaolin, blast-furnace-slag-based AAC composites were developed. The effects of storage conditions, specimen size and cyclic loading on the compressive strength were investigated, and the resulting figures were compared with the relevant values of classic binders. The strength (44.0 MPa) of the waste-based AAC composite significantly exceeds the required value (32.5 MPa) of clinker saving slag cement. Furthermore, following cyclic compressive loading, the residual strength of the waste-based AAC composite shows a slight increase rather than a decrease.

## 1. Introduction

Today, concrete is the most widely used material system in the world. The basic element of this is ordinary Portland cement (OPC), which has an annual global production volume of nearly 4 billion tons [1]. This amount is generating increasing environmental problems: suffice it to mention CO_2_ emission and the destruction of landscapes. For this reason, research for new types of binders is incessantly ongoing, and in the scope of such research alkali-activated cements (AACs), also termed geopolymers, play a leading role. However, it is expected that these non-cement-based binders will not completely replace conventional binders (e.g., OPC) but may be good alternatives. The great advantage of AAC production is that waste materials (such as power plant fly ash, slag) can also be used as raw materials. Furthermore, during the bonding process, AACs are able to bind other components considered as waste materials (e.g., red mud, demolition waste, glass waste, etc.). It is therefore important to continuously develop AACs, one possible way of which is to produce AAC-based composite material systems.

The history of alkali-activated systems dates back to the 1940s, with Feret being the first to try to produce non-cement-based binders using slag. The first major breakthrough was achieved in the late 1950s by Glukhovsky, who laid down the theoretical foundations of alkali cements. However, the exact description of AACs is attributed to Davidovits, who studied the structure of these material systems [2]. The most commonly used solid precursors for the production of geopolymers are metakaolin (MK) [3,4], ground granulated blast furnace slag (GGBFS) [5,6] and fly ash (FA) from coal combustion [7,8]. It is also noteworthy, however, that many other natural pozzolanic [9,10] and calcined materials [11,12] can also be alkali activated. Thanks to the use of these materials, CO_2_ emission becomes much lower than in the case of Portland cement. During the production of OPC, 44% of CO_2_ is released by the calcination of CaCO_3_. This is further exacerbated by the CO_2_ emissions from fuel consumption and the burning of fossil fuels for electricity generation [1]. Although the production of AACs does not require limestone, the production of alkaline activating solutions also involves CO_2_ emissions. In fact, the latter is estimated to account for 20% of CO_2_ emission from cement production [13]. In terms of total CO_2_ emission, Yang et al. conducted comprehensive research to analyze the manufacturing process of specimens involved in their study using OPC, GGBFS, FA and MK. Their experiments have shown that the higher the compressive strength value achieved, the higher the CO_2_ emission [14].

Depending on the chemical composition of the primary raw materials involved, AAC binders are usually classified into two main groups: one is the CaO-poor and the other is the CaO-rich system. Following Krivenko, Roy delineated the two classes using the formulas Me_2_O-Me_2_O_3_-SiO_2_-H_2_O and Me_2_O-MeO-Me_2_O_3_-SiO_2_-H_2_O [15]. The end products of the former group are alkaline zeolite-type minerals, while those of the latter are alkali-alkaline earth metal zeolites, calcium hydrosilicates and hydrocarbonates. Based on the reaction between the alkaline solution and the reactive components of the raw material, Palomo et al. also divided AAC binders into two groups [16]. In the case of the former, the Si and Ca (e.g., GGBFS) content reacts with the alkaline compound, while in the case of the second model, the Si and Al (e.g., MK or class F FA) content does the same. Accordingly, the main reaction product of the former process is the C-S-H gel (calcium-(aluminium)-silicate-hydrate) and that of the latter is a zeolite-like but concurrently amorphous polymer structure, which Davidovits collectively termed geopolymers [17].

The study of AACs raises a number of questions, mainly as far as the analysis of economic aspects is concerned. Therefore, a significant part of such research is seeking to find new opportunities of using special raw materials. Of course, the goal of such research is to use materials that are as cheap as possible, preferably considered waste, such as ground rubber from end-of-life tires, as well as construction and demolition materials, for example, bricks, concrete, glass and aerated concrete [18,19,20,21,22]. With the extensive development of different industries, the amount of industrial waste and by-products is increasing year by year, the recovery of which is still an unresolved issue. One of the great advantages of AACs is that, concerning these materials, wastes can be utilized not only as a filler [18,20,21,22] but also as a matrix [4,7,19]. The resulting composites may be suitable for use in special industrial processes. For example, rubber-containing AACs may be able to dampen vibrations that occur during the operation of larger machines (pumps, electrical equipment and motors, industrial and agricultural machinery).

As far as the global situation is concerned, almost 2 billion tires (~18 million tons) are produced each year worldwide, of which the share of the EU is ~5 million tons and the amount of rubber waste generated is ~3 million tons annually [23]. The primary problem is that there are few economically feasible solutions, so by today, sustainable and rational waste management has become one of the biggest global environmental issues. While storage in landfills dominated in the 1990s, today the focus is on energy recovery from rubber, and more importantly, on its use as a material [24]. The disposal of used tires is a major challenge due to their mixed composition [25]. One of the most advantageous ways of recycling entails crushing and grinding, after which the constituent components can be separated from each other through magnetic separation and screening. In addition to the production of rubberized asphalts [26] and shock-absorbing rubber tiles [27], the resulting rubber waste is also used in the construction industry to partially replace the aggregate (gravel, sand) in “classical” [18,28,29] and, in some cases, in AAC binder systems [30,31]. Based on its particle size, shredded, crumb and ground rubber waste [29] can be distinguished: shredded or cut pieces can partially replace coarse aggregate, crumb rubber can replace fine aggregate, while ground rubber may replace cement [18,29]. The size of the rubber particles used for production affects the mechanical properties of finished products.

In addition to the fact that rubber waste may reduce the workability and mechanical properties of AACs, an interesting question is how rubber affects the fire resistance of composites. Most studies agree that rubber-free AACs have better refractory properties than classical binders [32,33,34]. However, there are only few studies available that investigate the effect of elevated temperatures on rubber-containing AACs. Luhar et al. [34] prepared fly-ash-based AACs with 10 wt % rubber content, and the resulting composites were heat treated at different temperatures (200, 400, 600, 800 °C for two hours). They found that although the elevated temperature degrades the strength of the samples, the difference in percentage strength loss between the rubber-free control and the rubber-containing sample is not significant. Far-reaching conclusions cannot be drawn yet, but, probably, the small amount of rubber waste mixed into the AAC does not cause a drastic deterioration of the relevant properties.

It is obvious that the quality of the starting materials determines the properties of AAC composites, which—in some cases—have much lower strength values compared to “classical” binders [29,35]. Strength can be increased by dispersing different fibrous materials in the AAC matrix [36,37] and by improving the adhesion behaviour between rubber particles and the matrix [38,39,40]. The mechanical properties of fibre-reinforced AAC composites and of classical concretes lend themselves to comparison: the former can compete with the latter in terms of compressive and flexural strength, can even have better chemical resistance and abrasion resistance and can thus serve as a promising alternative. In this research, rubber-containing slag-based AAC composites were investigated: This paper therefore extends to the study of the relationship between mechanical properties (especially cyclic repetitive loading) and material structure characteristics, the applicability of kaolin-based fibrous materials for improving strength and the surface treatment of crumb rubber for improving wetting. Finally, in the scope of this study, a comparative experiment with traditional binders was performed, based on which the produced waste-based AAC composite can be deemed a promising building material.

## 2. Materials and Methods

### 2.1. Materials

The raw material for setting the boundary conditions for the model experiments was metakaolin (MK), which was obtained by calcining New Zealand kaolin (750 °C, 8 h). For waste-based AACs, ground granulated blast furnace slag (GGBFS) was used, which was provided by ISD DUNAFERR Zrt., Dunaújváros, Hungary. For the purpose of executing a comparative study with traditional binders, test specimens were prepared using CEM I 42.5 N type Portland cement and CEM III / B 32.5 N-LH / SR type blast furnace slag cement (produced by Duna-Dráva Cement Kft., Beremend, Hungary). The chemical and mineral compositions of MK and GGBFS are shown in Table 1 and Table 2: the former was determined by X-ray fluorescence spectrometry (XRFS) and the latter by X-ray diffraction (XRD).

The amount of each phase was determined with the help of Rietveld Analysis. Based on the XRD recording of the starting materials (Figure 1), it can be said that both materials contain a large amount of amorphous phase. In the case of MK, the crystalline phases are quartz and cristobalite, and, in the case of GGBFS, the main crystalline phase is merwinite. In addition, akermanite and bronwmillerite are also present as minor components.

The morphology of MK and GGBFS is shown in Figure 2: It is clearly visible that both materials have a heterogeneous particle size distribution. At the same time, the particles of MK are rounded, while the slag exhibits irregular particle shapes with edges. The particle size distribution of the raw materials and their median (D50) were determined with the help of a laser granulometer, MK: 14.7% < 2 μm, D50 = 23.93 μm; GGBFS: 13.41% < 2 μm, D50 = 13.84 μm.

The activating solution consisted in a mixture of analytical grade sodium hydroxide pellets (Reanal Laborvegyszer Kft., Budapest, Hungary) and a commercially available sodium silicate solution (ANDA Kft., Barcs, Hungary); the chemical composition of the latter was as follows: 28.6 wt % SiO_2_, 6.8 wt % Na_2_O and 64.6 wt % H_2_O. AAC mortars were prepared using standard quartz sand (CEN Standard Sand according to DIN EN 196-1, Normensand GmbH, Beckum, Germany) as a fine aggregate with a maximum particle size of 2 mm. In the case of AAC composites, in addition to the sand, crumb rubber from waste tires (Rubber Solution Kft., Budapest, Hungary) and kaolin wool fibres originating from the cutting waste from the thermal insulation of electric furnaces (Dualinvest Kft., Mosonmagyaróvár, Hungary) were also added. The maximum grain size of styrene butadiene rubber used in the scope of the experiments was 1 mm, and the average particle size was 500 μm; the kaolin wool fibres used were <10 mm long and 1–20 μm in diameter (Figure 3).

### 2.2. Sample Preparation

During the experiments, solid NaOH pellets were dissolved in waterglass (and in distilled water) for the preparation of the activating solution. For the production of the AAC specimens, an appropriately cooled activating solution was first added to the raw material (metakaolin or slag). During mixing, the same conditions were ensured, i.e., the slurry of the raw material–alkaline solution was mixed mechanically for 1 min at 900 rpm, and then the sand/crumb rubber/kaolin wool fibre as aggregate (with a 1:2 raw material: total aggregate mass ratio) was added to this mixture. The resulting mortar was further homogenized for 1 min in the case of sand and rubber and for an additional 2 min in the case of kaolin wool fibres.

The prepared fresh mortar was cast into 30 × 30 mm^2^ cylindrical PVC moulds and was cured under ambient conditions (21–23 °C and RH = 50 ± 10%) for 1 day before demoulding. After demoulding, the hardened samples were stored at ambient conditions until their testing at 7 and 28 days of age, respectively. In addition to the small samples, test specimens of dimensions 40 × 40 × 160 mm^3^ were also prepared in accordance with standard specifications used for cements, notably EN 12390-2.

In Stage 1, MK-based geopolymers were prepared using the following molar ratios: SiO_2_/Al_2_O_3_ = 3.6, Na_2_O/Al_2_O_3_ = 1.0; the mass ratio of sodium silicate and sodium hydroxide in the activating solution was 6.6 and that of the activating solution and metakaolin powder was 1.4. Standard sand used as filler was partially replaced with crumb rubber (5–45 wt %). GLENIUM C 300 type superplasticizer (G) was added to the matrix for better workability, and hydrophilizing agents—often used in the plastics industry—(succinic acid (SA), maleic anhydride (MA), xylene (X)) were added to improve the wettability of the rubber. The additives were mixed with the mortar in an amount of 0.5 wt % (based on the weight of the binder). To improve the strength of the prepared specimens, fibrous kaolin wool was added to the AAC matrix (0.5–3 wt %).

In Stage 2, the total amount of MK was exchanged for GGBFS. Thus, a mixture of the optimal composition determined in the first stage was prepared (10 wt % crumb rubber and 1 wt % kaolin wool fibre). The molar ratios used were as follows: SiO_2_/Al_2_O_3_ = 6.6, Na_2_O/Al_2_O_3_ = 1.1; the mass ratio of sodium silicate to sodium hydroxide in the activating solution was 6.6 and that of the activating solution and slag powder was 0.6. Subsequent to mixing, the surface activation of the rubber was performed. In this process, the rubber particles were subjected to chemical (concentrated H_2_SO_4_, NaOH, acetone) and physical (UV-A radiation, heat treatment at 80 °C) treatment before being incorporated into the AAC matrix. It is important to note that after acid and alkaline soaking, the rubber was washed with distilled water, so that neutral pH could be reached, and the rubber was then dried. The latter step also took place after surface modification in the organic solvent.

In Stage 3, the relevant physical properties of the waste-based AAC composite thus produced were compared with the relevant values of one commercially available classic Portland cement (CEM I 42.5 N) and one clinker-saving blast furnace slag cement (CEM III/B 32.5 N-LH/SR). During the tests, the effect of storage conditions (room temperature laboratory atmosphere and underwater storage, respectively) on strength of samples was investigated. The experimental parameters for each series are shown in Table 3, while the compositions of the solid fraction in Stage 3 are shown in Table 4.

### 2.3. Methods

The particle size distribution of the raw materials and their median (D50) expressed in μm were determined with the help of a Fritsch Laser Particle Sizer “Analysette 22“. Prior to the start of the tests, the suspensions of samples were treated in a water bath equipped with an ultrasonic stirrer and pump for 60 s in order to achieve proper dispersion and to eliminate intergranular aggregation.

Qualitative and quantitative phase analyses of the raw materials and the prepared AACs were carried out using a Philips PW 3710 X-ray diffractometer with CuKα (50 kV, 40 mA) radiation, 0.02° 2Θ/s speed (in the 2Θ range, 10–70°) and a graphite monochromator. The device control and data collection were performed using an X’Pert Data Collector. The X’Pert Highscore Plus software and ICDD PDF-2 reference database were used to evaluate X-ray diffractograms and to carry out Rietveld Analysis.

The strength properties of the AAC samples were characterized by their compressive and tensile strengths. A CONTROLS Automax5 device was used to perform related measurements. Prior to the start of the test, the surfaces of all specimens were polished flat and parallel. The tests were performed according to the relevant cement standard, namely EN 196-1, using a loading rate of 2400 N/s for compressive strength and 50 N/s for flexural strength tests at 7 and 28 days of age, respectively. In some cases, however, the sample sizes differed from the standard (ø30 × 30 mm cylindrical samples were also tested in addition to standard-size specimens). Concerning each mixture, three parallel measurements were made, and the 28-day average compressive strength values were reported.

The bulk density of the AAC composites was obtained as the quotient of the weight of the samples, and their volume was calculated based on their geometric size.

After testing the flexural strength of the standard size samples, two halves were obtained. Static compressive strength tests were performed on one of the two half pieces, while to the other half cyclic compressive loading tests were applied. For fatigue measurement, an Instron 5967 two-column tensile machine with an upper measuring limit of 30 kN in terms of load was used. The loading rate was chosen to be 2400 N/s (this is the same value as the one used for static compressive strength measurements), and the maximum load was selected to be one-third of the static compressive strength value obtained at 28 days of age. The number of cycles was 10,000.

Morphological studies were performed using a FEI/Thermo Fisher Apreo S scanning electron microscope (low vacuum mode) and a computer-controlled imaging system. The accelerating voltage was 20 kV for backscattered (BSE) and 10 kV for secondary (SE) electron imaging. The elemental composition of the samples was carried out using an EDAX AMETEK Octane Elect plus type energy dispersive X-ray analyzer using an accelerating voltage of 20 kV and a data collection time of 180 s.

An FT-IR analysis of untreated and treated rubber particles was performed using a Perkin Elmer Spectrum Two type instrument equipped with a platinum ATR adapter. During the measurements, the reflectance spectrum (wave number as a function of the Kubelka–Munk unit) was recorded by averaging 512 spectra.

According to ASTM D 2240 and ISO 868, the Shore A hardness of the rubber samples was determined with the help of a Mitutoyo HH-336-11 type digital durometer.

Computed tomography (CT) images of the produced binders were made using a Nikon XT H 225 ST type X-ray tomograph and the accompanying VG Studio 3.4 software. During the measurements, an accelerating voltage of 160 kV and a cathode current of 85 μA were used (1250 projections/recordings, 2 recordings per projection, 500 ms data collection time).

## 3. Results and Discussion

### 3.1. The Compressive Strength and Bulk Density of MK-Based AAC Samples

#### 3.1.1. The Effect of Rubber Content

In Stage 1, the MK-based AAC samples were prepared by partially replacing the standard sand used as filler with crumb rubber (see Table 3). As the amount of rubber increased, the mortars became more and more fragmentary and less workable. Figure 4a shows the compressive strength and bulk density values obtained, and Figure 4b,c summarize the results, which facilitated the setting of the boundary conditions (the effects of the additives and fibre reinforcement on the physical properties are shown in Figure 4b,c, respectively).

The relevant values of the control samples (with no crumb rubber) (25.5 MPa and 1925 kg/m^3^) strictly monotonically decrease as the amount of rubber increases. In Diagram (a), two steeper parts are observable: one after a rubber content of 10 wt % and the other after a rubber content of 35 wt %. Up to the first breaking point (ΔR = 10), there is a decrease of about 10% (7–9%) in strength, and this, in the middle section (ΔR = 15–35), is followed by a more drastic decrease of almost 60%. Finally, at ΔR = 45, only one-fifth (5.4 MPa) of the initial value is measured. Even if the AAC matrix is capable of binding large amounts of rubber waste and thus allowing for the production of lightweight building blocks (ρ_bulk_ ≤ 2000 kg/m^3^, σ_compressive_ ≥ 3 MPa), it is important to note that the aim of the present research is to develop a binder that can be used as a load-bearing substrate in the future. To achieve this, it is recommended to use 10 wt % of rubber waste. Although this amount is only a small part of the sand/rubber aggregate, its absolute volume is large. For the purpose of further experiments, Figure 5 clearly illustrates the volume difference between 10 wt % crumb rubber and sand of the same weight.

Even if the strength of the rubber particles is significantly lower than the strength of both the aggregate and the matrix, their amount mixed into the AAC system (10 wt %) does not cause a drastic deterioration of the properties relevant to the application scrutinised in the scope of this study. The relevant samples have a compressive strength of 23.3 MPa and a bulk density of 1779 kg/m^3^.

#### 3.1.2. The Effect of Additives

With respect to the AAC composite with 10 wt % rubber content, in an attempt to approximate and reach the strength value of the control samples (ΔR = 0, without additives: 25.5 MPa), attempts were made to improve the compatibility between the rubber and the AAC matrix through the use of additives. The application of rubber waste impairs the workability of AAC mortars, which was hoped to be improved by the addition of superplasticizer (G). In traditional binder systems (OPC), the superplasticizer can increase strength [37]. In order to investigate the effect of G on the AAC system, a rubber-free mixture containing GLENIUM C 300 (ΔR = 0, G) was prepared. Similarly to OPC, the additive has a favourable effect on the AAC system as well: a ~1.5-fold strength increase compared to the value of the initial control sample (from 25.5 to 38.0 MPa) was observed. However, contrary to all expectations, the superplasticizer did not improve the compressive strength of the samples containing rubber, and it even reduced strength by ~30% compared to the additive-free sample (ΔR = 10, without additives: 23.3 MPa and ΔR = 10, G: 15.6 MPa, respectively) (Figure 4b). However, in the case of cements and rubber-free AACs, the combination of dispersing and steric repulsion action of the polycarboxylate-ether-based superplasticizer improves workability and thus strength, the addition of rubber inhibits the mechanism of action triggered by the superplasticizer. Hydrophobic rubber particles repel the water-based activating solution during mixing; thus, air inclusions and pores are formed due to insufficient wetting. Through the investigation of the microstructure of the specimen containing both superplasticizer and crumb rubber (SEM analysis, Figure 6), it became clear that the wetting between the matrix and the rubber particles was insufficient, so the next goal in the scope of this study was to activate the surface of the rubber. Succinic acid (SA), maleic anhydride (MA) and xylene (X), which are frequently used and well-proven in the plastics industry, were used for this purpose. Based on Figure 4b, it can be concluded that in this case these hydrophilizing agents are not effective, and the use of all three additives leads to a decrease in strength (SA: −20%, MA: −15%, X: −30%).

#### 3.1.3. The Effect of Kaowool Fibre Content

Fibre reinforcement was used to increase strength. For this purpose, waste kaolin wool fibres were selected because their composition is close to the matrix composition. Si and Al ions are involved in the structure of both (SiO_2_/Al_2_O_3_ molar ratio 1.8 for kaowool and 2.5 for MK), thus it was presumed that satisfactory compatibility could be achieved. Initially, mixtures of different fibre contents were prepared without the addition of crumb rubber: The pertaining measurement results are shown in Figure 4c. The compressive strength values show a curve with one maximum point: the best result is obtained at a fibre content of 1 wt %, in which case a strength increase of ~35% compared to the control sample is observed (from 25.5 to 34.1 MPa). On the other hand, a further increase in the fibre content has a strength-reducing effect, which can be caused by the adhesion of the fibres and their inhomogeneous distribution in the matrix. The latter can be attributed to difficulties of fibre distribution associated with an increase in the amount of fibres. Ganesh and Muthukannan [36] experienced a similar phenomenon when producing high-performance glass-fibre-reinforced fly ash and slag-based AACs. In addition, the degradation of kaowool fibres in an alkaline medium emerged as a potential explanation. This assumption, however, was refuted experimentally: the fibres were kept in concentrated NaOH solution for 28 days, but no dissolution or weight loss was observed after the lapse of that period.

Considering the compositions giving the best results in the first series of experiments, an additive-free AAC mixture was prepared, in which 10 wt % of the aggregate was replaced by crumb rubber from waste tires and 1 wt % by kaolin wool fibres. The produced composite has a compressive strength of 28.2 MPa and a bulk density of 1802 kg/m^3^. It can be stated that the resulting decrease in strength caused by rubber doping can be compensated by using fibre reinforcement and as high as a 10% increase could be observed compared to the control sample (from 25.5 to 28.2 MPa).

### 3.2. The Investigation of GGBFS-Based AAC Samples

In Stage 2, the total amount of MK was replaced by GGBFS with a view to using industrial waste material as the raw material of AAC. For the sake of comparability, as a first step, a control sample was prepared in which the total amount of aggregate was standard sand, which was then substituted by rubber (5, 10, and 25 wt %, respectively). Subsequently, a mixture of the optimal composition determined in the first stage was prepared (ΔR = 10, ΔK = 1). The results are shown in Figure 7. The correlation between the rubber content and composite strength, which was observed when strength values could be kept at the same level upon adding certain amounts of rubber in the case of MK (Figure 4a), is not valid when GGBFS is used (Figure 7a). For the production of lightweight building elements, the maximum amount of rubber that can bind is 45 wt % in the case of MK and 25 wt % in the case of slag samples. The latter shows a ~90% decrease in strength compared to the control sample (from 40.6 to 5.4 MPa). The optimal 10 wt % rubber content for MK results in a ~60% strength drop in the case of GGBFS, which can be improved by fibre reinforcement (1 wt %). Based on the comparison of the compressive strength of the non-fibre-reinforced and fibre-reinforced samples, it can be stated that the use of kaowool, as in the case of MK, is also advantageous here: the values can be increased from 16.1 to 22.6 MPa. Although the percentage increase is higher (40%) than that observed in the case of the MK system (20%), the 22.6 MPa final value is lower than the 28.2 MPa optimum obtained in the scope of the first series of experiments. In this paper, hereinafter, this sample, which contains 10 wt % rubber and 1 wt % kaolin fibres, will be referred to as AAC UR.

The previously observed compatibility between matrix and crumb rubber is not adequate for slag-based AACs either. The problem caused by this situation can be remedied by treating the surface of the rubber particles [38,39,40]: This treatment can also improve strength values. In the scope of this study, the rubber particles were treated in five different ways: They were soaked in acid (H_2_SO_4_), alkali (NaOH) and organic solvent (acetone), as well as received UV irradiation and heat treatment at 80 °C, respectively. The relevant results are shown in Figure 7b. Compared to the untreated sample, all methods proved to be effective, but the strength of the specimens containing sulfuric acid-treated rubber particles was the highest. In this case, the initial value was improved by almost 55% (from 16.1 to 24.8 MPa), while a smaller increase of 25–40% could be achieved through the application of the other treatment techniques. Based on the investigation of the morphology of the specimens (Figure 8), it can be stated that the adhesion between the crumb rubber and the matrix can be promoted through different treatment methods, which basically determines the behaviour of the composite.

For finding an explanation to the physical changes experienced when differently treated rubber particles were used, FT-IR studies were performed (Figure 9). Colom et al. [39] reported that treatments can in some cases lead to the formation of new functional groups or to the disappearance of old ones. It has become clear that each treatment used in the present research not only physically affects the properties of the rubber waste but also causes significant transformations in chemical bonds. The emerging C=O, C–O and C=C bonds (at 1730–1690, 1150–1080 and 880–840 cm^−1^, respectively) contribute to increasing the strength of the modified rubber particles, which may help to improve the mechanical properties of the resulting composite. To demonstrate this, a Shore A hardness test of untreated and treated tires was performed: The results of this test are shown in Table 5.

The hardness of all treated rubber samples increased compared to the untreated sample, and the most significant change was observed in the case of the sulfuric acid treatment (11.5%). This result is in agreement with the results obtained in the compressive strength test of composites.

For proving that fibre reinforcement is effective even in the case of the use of sulfuric acid-treated rubber particles (ΔR = 10), a sample containing 1 wt % kaowool (ΔK = 1) was prepared. The compressive strength of the relevant samples is 33.5 MPa, which represents an increase of 35% compared to the strength of the sample without fibre reinforcement (24.8 MPa) and of almost 50% compared to the value of AAC UR composite (22.6 MPa). Furthermore, it is ~20% higher than the optimum value of the first series (28.2 MPa). Hereafter, in the scope of this study this sample will be referred to as AAC TR.

### 3.3. The Results of Comparative Tests

In Stage 3, the physical properties of the GGBFS-based systems (pure slag, AAC UR and AAC TR) were compared with the relevant values of traditional binders (CEM I 42.5 N type Portland cement, in short: CEM I; and CEM III/B 32.5 N-LH/SR type slag cement, in short: CEM III). During the test, the same sample preparation, storage and test conditions were ensured. As there are no applicable standard specifications for AACs, all specimens were first stored in air at room temperature after demoulding. It can be clearly seen in Figure 10a that under such conditions, waste material-based samples show better strength properties than the other two cement-based binders.

However, it should be noted that, over time, a number of hairline cracks formed on the surface of traditional binders, which are most likely caused by the dehydration process observable on the surface. So that the nature of the cracks could be determined (only on the surface or within the whole structure?), a 3D examination of the samples was performed using X-ray tomography (CT). The internal structure of the samples was investigated after reconstruction. The resulting Figure 11 clearly shows that the cracks formed on the surface penetrate into the inside of the samples and also appear around the pores present in the structure. Both pores and cracks have a strength-reducing effect, which explains a deviation from the standard value (17.2 MPa instead of 42.5 MPa in the case of CEM I and 17.3 instead of 32.5 MPa in the case of CEM III).

If the samples were stored under water in accordance with the requirements of standard EN 12390-2, the moisture content was sufficient for both the bonding and solidification mechanisms, thus the tendency to crack could be eliminated. In the case of the slag-based AAC system, said cracking problem was not observed. Nevertheless, the remaining question was what result would be obtained if the specimens were stored under standard conditions as the ones applicable to cement mortars. It can be seen in Figure 10a that it is advantageous to keep the samples under water in the case of all binder systems. Regardless of the method of storage, the strength values obtained show a similar trend to that of air-stored samples. The strength of the AAC TR sample can be increased by 15% under the new conditions detailed above (from 33.5 to 38.3 MPa). From this, it is clear that underwater storage is also beneficial for AACs.

Still, CEM I specimens do not reach the value of the second strength class (42.5 MPa) specified in the product standard (EN 197-1). However, it should be noted that this requirement applies to samples prepared in the standard way and size (40 × 40 × 160 mm^3^ column). As the experiments carried out so far were performed using small pieces (ø30 × 30 mm cylinder), in the scope of the study, it was considered important to perform a size increase also in the case of the waste-based AAC system. The results of the tests are shown in Figure 10b. The strength of all compositions improved when the standard size was used: CEM I was the most prominent, and the other four systems showed roughly the same results (15–20%). At this point, it is important to note that, using only waste materials, a new composition (AAC TR) has been developed that shows a compressive strength (44.0 MPa) exceeding the strength of CEM III clinker-saving green cement (standard value: 32.5 MPa, actually achieved value: 42.9 MPa). Looking at the relative proportions of the solid components of the AAC TR and CEM III binder systems (Figure 12), it can be seen that while the waste content is below 20 wt % for slag cement mortar, it exceeds 40 wt % in the case of AAC composite. Furthermore, by using crumb rubber and kaowool fibres concurrently, the amount of quartz sand used as aggregate can be reduced by ~15 wt %.

In the last phase of this comparative study, samples CEM I, CEM III and AAC TR were subjected to cyclic loading for the examination of the effect of fatigue on residual strength. First, the flexural strength of the samples prepared in the standard size was measured. Then, one of the two halves formed after the measurement was subjected to a static compressive strength test, while the other was subjected to a cyclic compressive loading test (loading rate: 2400 N/s; maximum load: one-third of the static compressive strength value; number of cycles: 10,000). After the cyclic loading test, the static compressive strength of these pieces was measured; the results are shown in Figure 13.

After cyclic loading, the residual strength of the CEM I samples decreased by ~10% compared to their static compressive strength, while that of the other two binder systems increased slightly (5% and 8%, respectively). The different results can be attributed to differences in the initial components. Both the slag used as a binder and the rubber particles used as an aggregate can improve the final product’s resistance to cyclically repetitive mechanical effects. The former contributes to this through a combination of pozzolanic activity and the filler effect due to particle size, while the latter through elastic properties. In the case of OPC, crumb rubber from waste tires is often used to increase the resistance to dynamic loads, impact resistance, deformation capacity and fatigue life [18]. However, there are only few sources available that show the development of compressive strength values after a given fatigue cycle. Most studies agree that rubber particles are able to absorb some of the energy generated during loading to prevent the propagation of cracks generated during load cycles and to improve bonding between crack surfaces [18,29,35]. However, no clear conclusion can be drawn as to how the bond between the rubber particles and the matrix changes as a result of repeated compressive loading. Huang et al. [35] reported a decrease in strength in the case of all OPCs and fly-ash-based, rubber-free and rubber-containing samples: the drop in strength was greater in the case of samples containing rubber. These different results were attributed to the low strength of the rubber particles and the effect of cavity formation due to the hydrophobic nature of the materials involved. In the scope of the present research, the problems mentioned were remedied by sulfuric acid treatment of the AAC TR samples (Figure 8b, Table 5), but this only partially explains the improvement in residual strength after cyclic loading. Our empirical experience has shown that styrene butadiene rubber hardens under repeated compressive stress, thereby increasing its strength. When this material is used in AAC systems, presumably, this very process takes place, which thus contributes to the improvement of the mechanical properties of the resulting composite.

It can be clearly seen that the AAC composite containing rubber and kaowool fibres shows satisfactory physical properties, which is reflected not only in its strength values but also in its resistance to cyclic loading. These properties may make the AAC composite suitable for use as a structural material in those places that are subject not only to static but also to repetitive loading. Thus, it can be claimed that, with the help of our studies, it has been proved possible to produce a waste-based composite material system whose mechanical properties are competitive with those of traditional binders.

## 4. Conclusions

This paper aimed to prepare and investigate AAC composites with industrial waste materials as the majority of their solid fraction. The primary goal of the study was to find the optimal crumb rubber/sand ratio that does not cause a drastic deterioration of the properties relevant to the application of AAC composites. Based on the experiments performed, the following conclusions can be drawn:The standard sand used as filler can be partially replaced by crumb rubber from waste tires: up to 45 wt % in the case of a pure MK system and up to 25 wt % in the case of GGBFS dotted with impurities. Any further increase in the amount of rubber causes a large deterioration of physical properties.Wetting between the rubber particles and the matrix is inadequate, which could not be remedied either by the superplasticizer commonly used in the cement industry or by hydrophilizing agents proven in the plastics industry. When these additives were used, there was a 15–30% decrease rather than an expected increase in strength.The decrease in strength due to rubber particles can be compensated for through the use of fibrous kaolin wool waste. The best result is obtained with the addition of 1 wt % fibre content: given this, in the case of an MK system, a ~35% strength increase is observed, while in the case of GGBFS, a ~40% strength increase is observed.The adhesion between the rubber particles and the matrix can be improved by chemical or physical surface treatment of the rubber. The increase in strength due to the change in the composition of the rubber waste was proven by the Shore A test. The most significant increase (~55%) can be achieved through sulfuric acid treatment: in this case the rubber particles adhere to the matrix the most extensively, and no cracks are visible on the contact surfaces.The addition of kaolin wool fibres also improves the compressive strength values when sulfuric-acid-treated crumb rubber is used. This way, an increase of almost 50% compared to the sample containing untreated rubber and an increase of ~20% compared to the optimum of the MK system can be achieved.When stored at ambient conditions—unlike cement-based specimens—no dehydration process takes place on the surface of waste-based composites. Through using underwater storage, the strength of the AAC TR sample considered optimal can be increased by 15%. It is clear from this that storage under standard conditions in the case of cement mortars is also advantageous for AACs.Strength values can be maximized by producing specimens in a standard way and size, i.e., by scale-up experiments. The strength available through using wastes is 44.0 MPa, which exceeds the standard value for clinker-saving cement.The flexibility of binders can be increased by the addition of rubber, which thereby also improves the binders’ resistance to cyclically repetitive mechanical effects. The post-fatigue residual strength of the waste-based composite (AAC TR) produced is jointly determined by the slag responsible for the pozzolanic activity and the added rubber particles.

On the whole, it can be claimed that if experimental parameters are properly selected, higher value-added AAC composites can be produced using waste materials. The physical properties of such materials are competitive with those of traditional binders. Moreover, the compressive strength of the AAC TR sample, which contains 2.5 times more waste than the mortar produced from the CEM III slag cement, exceeds the strength value stipulated by the standard for clinker-saving green cement.

## Figures and Tables

**Figure 1 materials-14-05815-f001:**
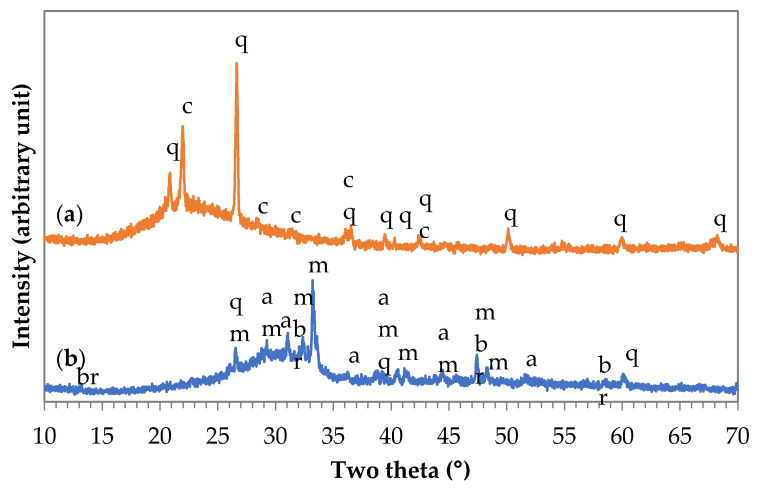
XRD patterns of MK (**a**) and GGBFS (**b**) (q: quartz PDF 33-1161, c: cristobalite PDF 39-1425, m: merwinite PDF 35-0591, a: akermanite PDF 01-087-0046, br: brownmillerite PDF 01-074-3673).

**Figure 2 materials-14-05815-f002:**
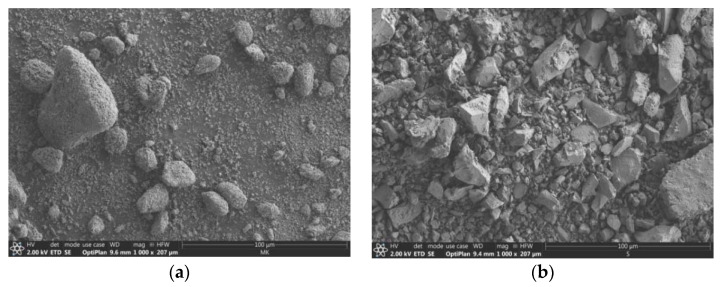
SEM micrograph of MK (**a**) and GGBFS (**b**).

**Figure 3 materials-14-05815-f003:**
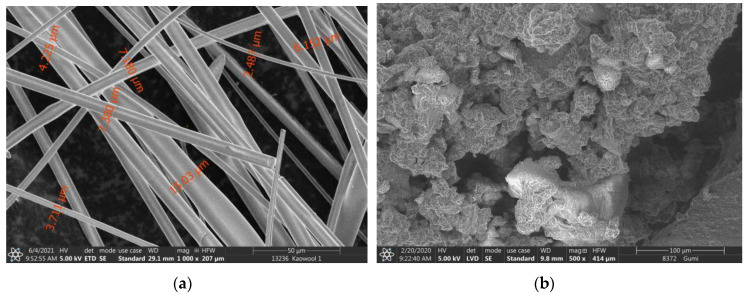
SEM micrograph of kaowool fibres (**a**) and crumb rubber (**b**).

**Figure 4 materials-14-05815-f004:**
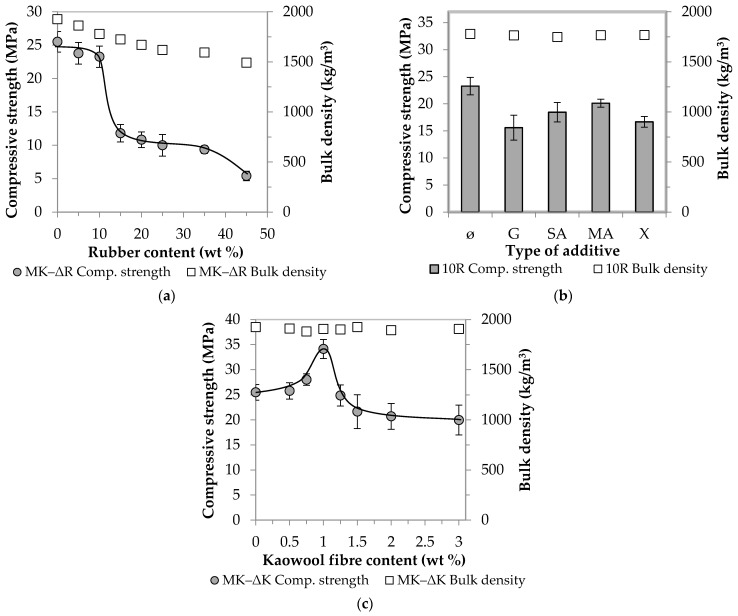
Effect of rubber content (**a**), additives (**b**) and fibre reinforcement (**c**) on compressive strength and bulk density of MK-based AAC samples.

**Figure 5 materials-14-05815-f005:**
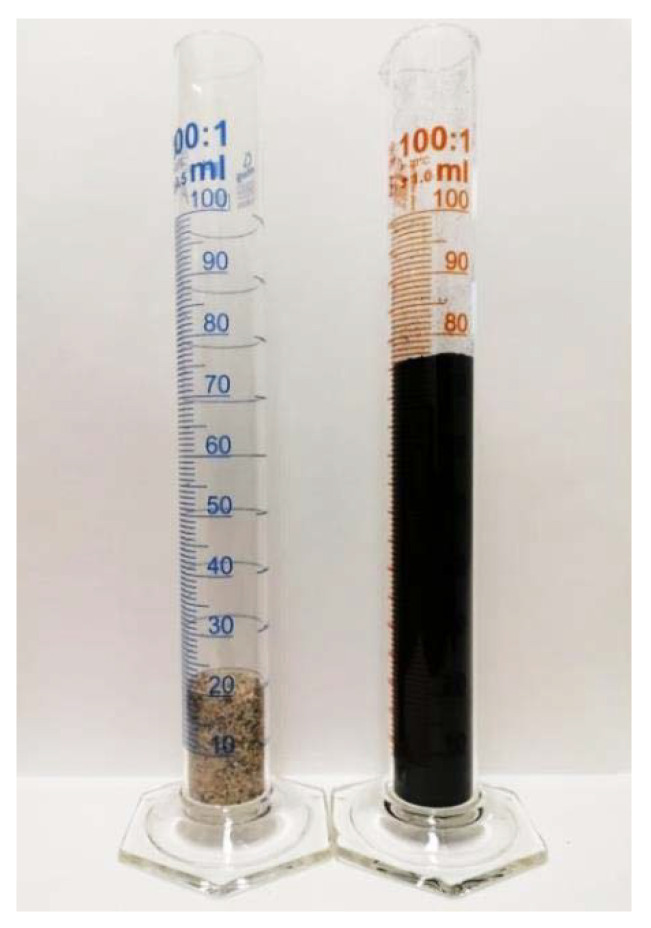
Volume difference between sand (left) and crumb rubber (right) of the same weight (10 wt %).

**Figure 6 materials-14-05815-f006:**
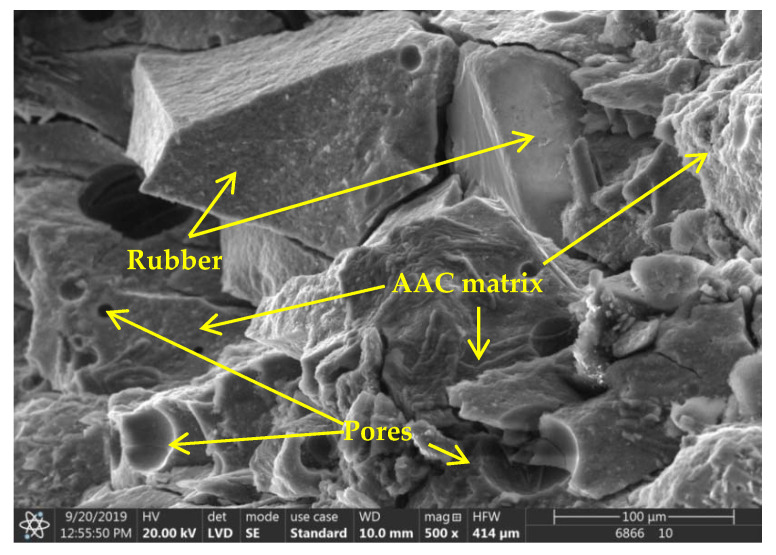
SEM micrograph of rubber doped (10 wt %) MK-based AAC sample.

**Figure 7 materials-14-05815-f007:**
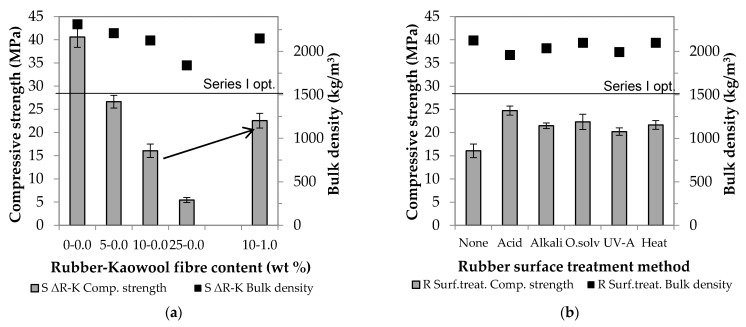
Effect of rubber–kaowool fibre content (**a**) and rubber surface treatment method (**b**) on compressive strength and bulk density of GGBFS–based AAC samples.

**Figure 8 materials-14-05815-f008:**
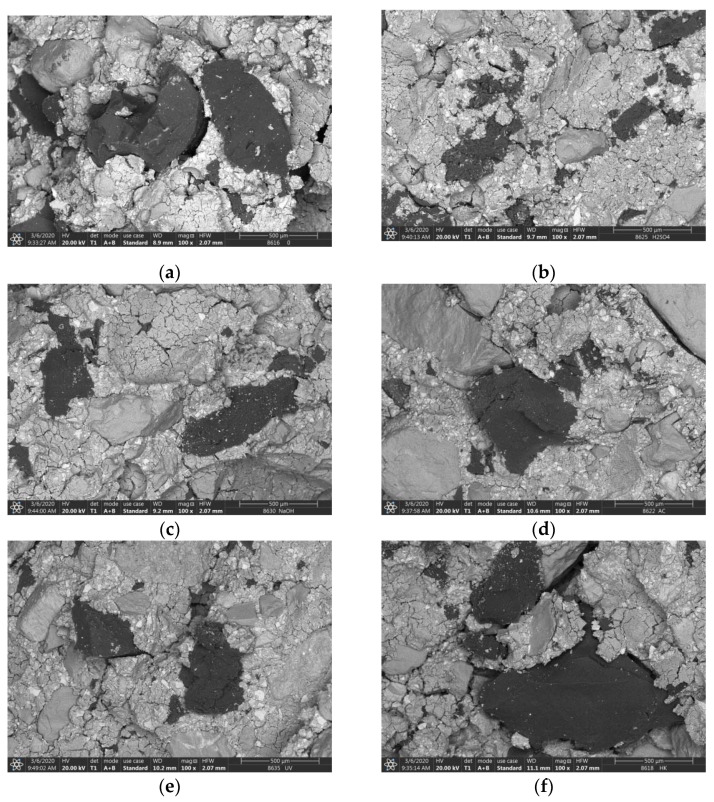
Adhesion of differently treated rubber and matrix ((**a**): untreated, (**b**): H_2_SO_4_, (**c**): NaOH, (**d**): acetone, (**e**): UV-A radiation and (**f**): heat treated).

**Figure 9 materials-14-05815-f009:**
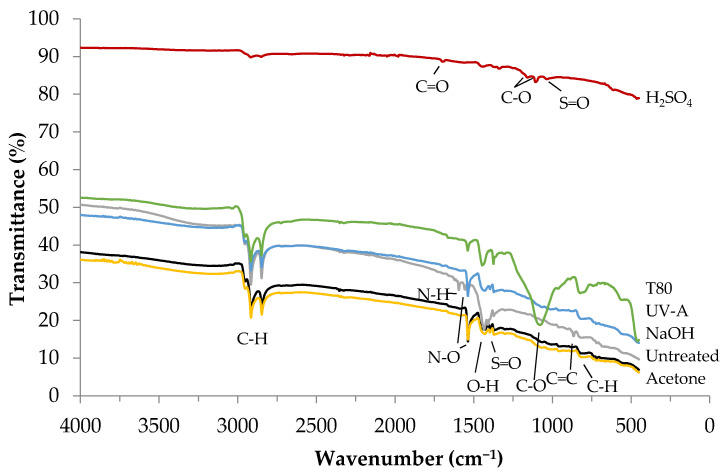
FT−IR spectra of untreated and treated crumb rubber.

**Figure 10 materials-14-05815-f010:**
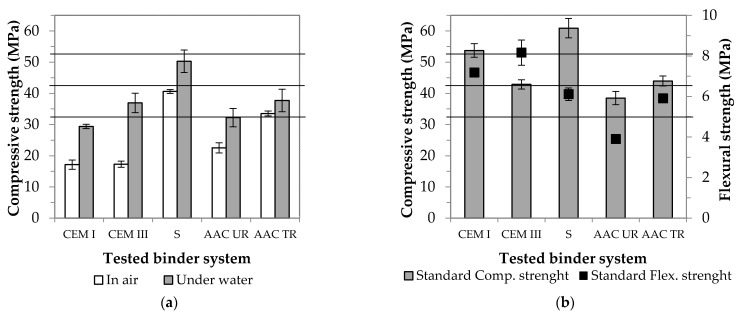
Effect of storage conditions (**a**) and size of specimens (**b**) on strength (Horizontal lines indicate strength classes according to the EN 197-1 standard.).

**Figure 11 materials-14-05815-f011:**
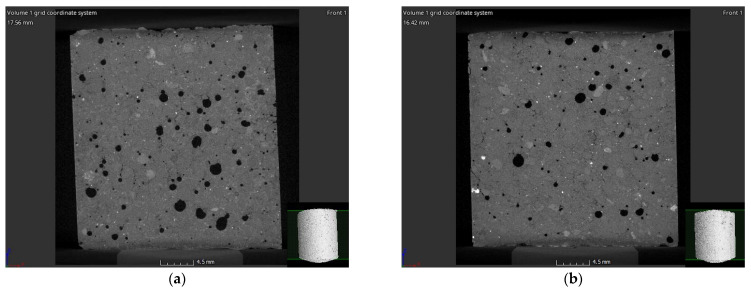
Surface and internal cracks in the case of CEM I (**a**) and CEM III (**b**) samples.

**Figure 12 materials-14-05815-f012:**
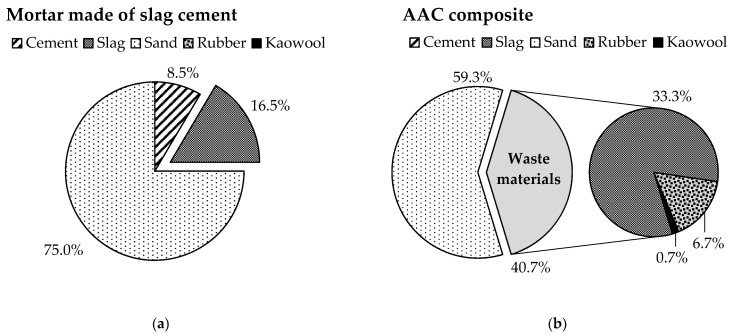
Compositions of the solid fraction of slag cement mortar (CEM III) (**a**) and AAC composite (**b**).

**Figure 13 materials-14-05815-f013:**
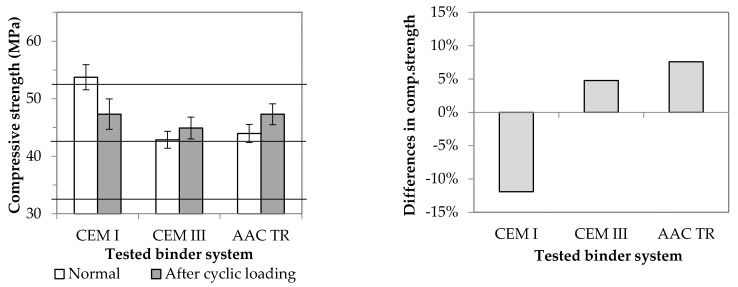
Difference between normal and post−cyclic compressive strength (Horizontal lines indicate strength classes according to the EN 197−1 standard.).

**Table 1 materials-14-05815-t001:** Chemical composition (wt %) of Metakaolin (MK) and Slag (GGBFS).

	SiO_2_	Al_2_O_3_	TiO_2_	Fe_2_O_3_	CaO	MgO	Na_2_O	K_2_O	SO_3_	LOI
MK	56.26	38.04	0.17	0.88	0.54	0.47	0.42	0.95	-	2.16
GGBFS	34.05	6.45	-	-	47.36	8.06	1.08	0.57	1.48	0.95

**Table 2 materials-14-05815-t002:** Phase composition of Metakaolin (MK) and Slag (GGBFS).

	Quartz	Cristobalite	Merwinite	Akermanite	Brownmillerite	Vitreous Phase
MK	4.6	3	-	-	-	92.4
GGBFS	0.2	-	11.1	1.2	0.6	86.9

**Table 3 materials-14-05815-t003:** Experimental parameters.

Series	Starting Material	Amount, wt %		Additive	Surface Treatment of Rubber	Storage Conditions	Sample Size (mm)
		Rubber (ΔR)	Kaowool (ΔK)				
1	MK	0, 5, 10, 15, 20, 25, 35, 45	0.0	-	-	T_r_ atm	ø30 × 30
		10	0.0	G, SA, MA, X			
		0.0	0.0, 0.5, 0.75, 1.0, 1.25, 1.5	-			
2	GGBFS	0, 5, 10, 25	0.0, 1.0	-	-	T_r_ atm	ø30 × 30
		10	0.0	-	acid, alkali, organic solvent, UV-A, T_80_		
3	GGBFS CEM I CEM III/B	10	1.0	-	acid	T_r_ atm, T_20_ water	ø30 × 30, 40 × 10 × 160
		-	-	-	-		

Additives: GLENIUM C 300 type superplasticizer (G), succinic acid (SA), maleic anhydride (MA), xylene (X); T_r_ atm: curing under ambient conditions (21–23 °C and 50 ± 10% RH) until testing after demoulding; acid: treatment with sulphuric acid (96 wt %) for 1 min, based on research settings applied by Colom et al. [39]; alkali: soaking in sodium hydroxide solution (50 wt %) for 1 h; organic solvent: soaking in acetone solution (50 vol%) for 1 h; UV-A: treatment with UV-A (380 nm) irradiation for 72 h; T_80_: heat treatment at 80 °C for 24 h; T_20_ water: storage at 20 ± 2 °C under water until testing after demoulding.

**Table 4 materials-14-05815-t004:** Compositions of solid fraction of third series.

Amount, wt %	CEM I 42.5 N	CEM III/B 32.5 N-LH/SR	AAC Composite
Cement	25	8.5	-
Slag	0	16.5	33.3
Sand	75	75	59.3
Rubber	-	-	6.7
Kaowool	-	-	0.7

**Table 5 materials-14-05815-t005:** Shore A hardness of untreated and treated crumb rubber samples.

Surface Treatment of Rubber	Untreated	H_2_SO_4_	NaOH	Acetone	UV-A Radiation	Heat Treatment
Shore A hardness	49.1	54.7	50.1	51.3	51.1	53.0
Change (%)	-	11.5	2.1	4.5	4.2	8.0

## Data Availability

Data is contained within the article.

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
