# Peer review of "Development of Waste-Based Alkali-Activated Cement Composites"

_materials, 2021, doi:10.3390/ma14195815_

Round 1

Reviewer 1 Report

This paper reports data on the study of the relationship between mechanical and material structure characteristics, the applicability of kaolin-based fibrous materials for improving strength and the surface  treatment of crumb rubber for improving wetting.

In general all the experiments seem to have been carefully conducted, but in my opinion some data need to be better discussed and the authors need to better define the objectives of their work, which in my view is a little confused, it is not clear where they intend to go.

  • Did the authors carry out any treatment or purification of starting material? in my opinion they should present the kaolinite x-ray data.
  • In my opinion the authors could further discuss the data obtained from x-rays, for example an increase in the amount of amorphous silica in the material is visible after heat treatment.
  • What caused this increase in amorphous silica, that is normally observed by acid treatments on kaolinites which cause partial dissolution of Al3+ cations from the octahedral sheet?
  • In my opinion the calculation of the surface area of the starting material and material after heat treatment can be an important factor in the characterization.
  • SEM data could be better complemented with particle size distribution.
  • Wouldn't the results obtained by impregnating rubber with kaolinite be sufficient justification for not using metakaolin, since there is a high energy expenditure to obtain it?
  • From my point of view the authors do not make clear to the reader the importance of using kaolin and metakaolin, the authors need a better conclusion for their data.

Author Response

Dear Reviewer,

Thank you for your constructive comments and suggestions. I would like to answer your questions and comments in summary.

In our study, metakaolin obtained from heat treatment by kaolin was used only as a model material to optimize the experimental parameters. The basic goal of the research was to produce alkali activated cement (AAC) composites whose starting components are mainly waste materials (ground granulated blast furnace slag, crumb rubber and kaolin wool fibres). However, the slag selected as raw material of AAC composites is dotted with impurities and cannot be considered as a pure material.

Kaolin is an expensive raw material and, in addition, is non-reactive in its original form and therefore cannot be used as a starting component for AACs. To achieve sufficient reactivity, it must be activated thermally or mechanochemically (by high-energy intensive milling).  Previously we have dealt with this area in more detail, the article below describes our experience: Balczár, I., Korim, T., Kovács, A., Makó, É.: Mechanochemical and thermal activation of kaolin for manufacturing geopolymer mortars – Comparative study; https://doi.org/10.1016/j.ceramint.2016.06.182

With both activation methods, the production of reactive materials is energy and time consuming, but cannot be avoided. AACs with similar strengths can be produced by the two methods, heat treatment was chosen in the present study. After heat treatment, an increase in the amorphous silica content is indeed observed as a result of the reactions that take place during combustion. Kaolinite clay mineral loses its structural water content between 550-600 °C; the dehydroxylation temperature depends on the purity and degree of crystallization of kaolin.

Si4O10(OH)8Al4 → 2(Al2O3·2SiO2) + 4H2O

The formed metakaolinite has an amorphous structure, the unique feature of which is that in addition to the tetrahedral silicon ion, in addition to the original six coordination, aluminium ions with coordination 5 and 4 also appear during heat treatment, which further increases the reactivity of metakaolin.

In the study, we were considered important to present the XRD curve of metakaolin due to the characterization of the starting material. Kaolin in its original form was not used in our experiments, so we did not include its XRD curve in the manuscript. However, if the reviewer feels necessary, we will of course insert it. The specific surface area values of kaolin and metakaolin are also available, however, no significant change was observed after heat treatment (kaolin: 19.4 m2/g, metakaolin: 18.0 m2/g). The particle size distribution curves of the starting materials (metakaolin and slag) exist, but were not included in the manuscript due to space savings; if you feel the need, we will of course insert it in the right place.

I would like to briefly summarize the importance of metakaolin in the study. Kaolin should not be used instead of metakaolin because it would not be able to react with the activating solution without thermal or mechanochemical activation. Metakaolin was used as a model material to optimize the experimental parameters.

I would like to briefly summarize the basic goal of the study. After recording the experimental parameters, the starting material of the AACs was completely replaced with slag formed as an industrial by-product to produce waste-based composites. By incorporating rubber waste into the AAC matrix, higher value-added composites can be produced that are competitive with classical binders in terms of their properties. In the manuscript, the normal compressive strength and the residual strength after fatigue testing of the waste-based AAC composite were compared with the relevant values of concrete made from two commercially available cements.

Sincerely, 

Adrienn Boros

Reviewer 2 Report

1. The Introduction section needs to be revised, a clear and logical expression is significant in this part. However, the existing section is kind of confusing.

2. In page 8, line 276, the expression "a rubber content  of 10" needs to be changed as “a rebbur content of 10 wt%”

3. This work is generally readable, however, it needs to be thoroughly refined, due to some expression makes no sense.

4. I do believe that more characterization methods are essential for this work to further verify the product system of samples with waste incorporation. TG and XRD analyses are expected to be added in this work, and , some articles may be helpful:

1) Ohmic heating curing of high content fly ash blended cement-based composites towards sustainable green construction materials used in severe cold region

2) Influence of mineral admixtures on carbonation curing of cement paste

3) Multi-structural evolution of conductive reactive powder concrete manufactured by enhanced ohmic heating curing

5. The Conclusion part is too lengthy, only necessary and important conclustions are required to be listed in this section.

Author Response

Dear Reviewer,

Thank you for your constructive comments and suggestions. I would like to answer your questions and comments point by point.

  • Comment 1: The Introduction section needs to be revised, a clear and logical expression is significant in this part. However, the existing section is kind of confusing.

Response: We have revised the Introduction part, in the first paragraph we wrote in more detail about new, less environmentally damaging alkali activated cements to emphasize that AACs are able to function as alternative binders without the addition of cement.

In the Introduction part, we tried to provide sufficient background about AACs (their significance, history, starting material, structure, environmental impact). The following paragraphs followed the logical line of thought shown below: The great advantage of AACs is that they can use industrial by-products and wastes as raw materials and can also bind wastes in their matrix, making the production process of AAC cheaper. One such waste may be crumb rubber from end of life tires, the amount of which is increasing year by year, but recycling is still an unresolved issue. When mixed rubber in AACs, it reduces the strength of the composite, but this can be compensated by using fibre reinforcement.

We hope that this will make the structure of the literary background more understandable. If there is any confusing section left, please let us know which one it is and we will try to change it.

  • Comment 2: In page 8, line 276, the expression "a rubber content  of 10" needs to be changed as “a rubber content of 10 wt%”

Response: Thank you for pointing this out. I agree with this comment, I have added "wt%" to this sentence.

  • Comment 3: This work is generally readable, however, it needs to be thoroughly refined, due to some expression makes no sense.

Response: I'm not sure I understand exactly what you mean here.  Would you mind sending me examples of what are expressions that don’t make sense? I highly appreciate your help.

  • Comment 4: I do believe that more characterization methods are essential for this work to further verify the product system of samples with waste incorporation. TG and XRD analyses are expected to be added in this work, and, some articles may be helpful:

1) Ohmic heating curing of high content fly ash blended cement-based composites towards sustainable green construction materials used in severe cold region

2) Influence of mineral admixtures on carbonation curing of cement paste

3) Multi-structural evolution of conductive reactive powder concrete manufactured by enhanced ohmic heating curing

Response: Thank you for this suggestion. It would have been interesting to explore this aspect. However, in the case of our study, it seems slightly out of scope because the problems that arise in the case of classical binders (e.g., Ordinary Portland Cement - OPC) may not be relevant for alkali activated cements (AAC). AACs do not contain cement; these new types of non-cement based binders differ from PCs in many ways.

For example, the AACs

- are built on a different raw material base than the “classic” binders known so far,

- are able to bind and solidify in different ways, no water is required for this process,

- do not generate heat,

-’s reaction products are not hydrate products,

- behave differently against external influences, etc.

In the articles you suggest, waste materials are only added as additives in the cement-based binder system, in contrast, the aim of our study is to make a binder, in which raw material is completely waste (slag) and a part of fine aggregate is also waste (crumb rubber from end of life tires). By incorporating rubber waste into the AAC matrix, higher value-added composites can be produced that are competitive with classical binders in terms of their properties.

The aim of our study was not to determine the phase composition of the produced AAC composite (XRD analysis is limited because the AAC composite has a high sand content; the high intensity of the quartz peaks in the sand suppresses the lower intensity peaks of reaction products). We were looking for answers to the following questions:

- is able the waste rubber to bind to the AAC matrix,

- how the loss of strength caused by the rubber can be compensated,

- what is the value of the normal compressive strength and the residual strength after cyclic loading (after 10000 cycles) of the AAC composite containing both rubber and slag,

- are the strength values of the AAC composite comparable to the relevant values of classical binders?

In the case of the last question the CEM I 42.5 N type Portland cement and CEM III/B 32.5 N-LH/SR type blast furnace slag cement used for the comparative studies were only references of classical binders.

  • Comment 5: The Conclusion part is too lengthy, only necessary and important conclustions are required to be listed in this section.

Response: We have, accordingly, modified the Conclusion part to emphasize this point.

In addition to the above comments, all spelling and grammatical errors pointed out by the reviewers have been corrected. We look forward to hearing from you in due time regarding our submission and to respond to any further questions and comments you may have.

Sincerely,

Adrienn Boros

Reviewer 3 Report

This work studied the performance of alkali-activated slag/metakaolin containing rubber. Many experimental methods have been conducted to evaluate the physical and mechanical properties of the composites. XRD, DTA, and SEM are the main instrumental analyses which investigate the phase composition of the prepared composites. The English language is OK and the representation of results is very good. Some drawbacks should be fixed before the acceptance of this paper.

  • The results should be discussed in the light of the previously published works.
  • SEM should be provided by EDS analysis.
  • What about the thermal conductivity of composites?

Please add a paragraph on the firing resistivity of composites having rubber.

Author Response

Dear Reviewer,

Thank you for your constructive comments and suggestions. I would like to answer your questions and comments point by point.

  • Comment 1: The results should be discussed in the light of the previously published works.

Response: Thank you for pointing this out. We agree with this comment. We tried to discuss our results in this study in the light of the previously published relevant works.

  • Comment 2: SEM should be provided by EDS analysis.

Response: The results of the EDS analysis are available, but were not included in the manuscript due to space savings; if you feel the need, we will of course insert it in the right place. Would you mind sending me examples of which SEM images should be provided by EDS analysis? I highly appreciate your help.

  • Comment 3: What about the thermal conductivity of composites?

Response: The thermal conductivity of the relevant samples are available. Heat conductivity was measured using the MTPS (Modified Transient Plane Source) method with a C-Therm TCi equipment. Before characterization, the bottom of the samples was polished and the powder removed from the exposed pores; no contact agent was used for measurement. The MTPS technique is a fast thermal conductivity measuring method and does not require large-sized specimens, but the thermal conductivity value obtained is lower than the one measured with a heat flow meter. The values of the thermal conductivity of our samples measured according to the MTPS method are as follows:

Blast furnace slag based alkali activated cement (S): 0,165 W/mK,

AAC sample which contains 10 wt% acid treated rubber and 1 wt% kaolin fibres (AAC TR): 0,168 W/mK,

CEM I 42.5 N type Portland cement (CEM I): 0,165 W/mK,

CEM III/B 32.5 N-LH/SR type blast furnace slag cement (CEM III/B): 0,160 W/mK.

We did not consider it important to present these values because our manuscript mainly focused on the evolution of compressive strength. If you feel the need, we will of course insert the thermal conductivity of relevant samples in the paper.

  • Comment 4: Please add a paragraph on the firing resistivity of composites having rubber.

Response: Thank you for this suggestion. We have revised the Introduction part and wrote a paragraph about fire resistance of rubber-containing AACs. It would be interesting to explore the effect of extreme temperature conditions on strength in the future.

In addition to the above comments, all spelling and grammatical errors pointed out by you have been corrected. We look forward to hearing from you in due time regarding our submission and to respond to any further questions and comments you may have.

Sincerely,

Adrienn Boros

Round 2

Reviewer 3 Report

The authors have carefully revised the manuscript. This draft is acceptable in the current for. .